# A Symbiotic Meal Containing Extruded Sorghum and Probiotic (*Bifidobacterium longum*) Ameliorated Intestinal Health Markers in Individuals with Chronic Kidney Disease: A Secondary Analysis of a Subsample from a Previous Randomized and Controlled Clinical Trial

**DOI:** 10.3390/nu16121852

**Published:** 2024-06-13

**Authors:** Haira Guedes Lúcio, Rita de Cassia Stampini Oliveira Lopes, Mariana Juste Contin Gomes, Alessandra da Silva, Mariana Grancieri, Ceres Mattos Della Lucia, Valéria Aparecida Vieira Queiroz, Bárbara Pereira da Silva, Hercia Stampini Duarte Martino

**Affiliations:** 1Nutrition and Health Department, Federal University of Viçosa, Campus Universitário, Av. Purdue, s/n, Viçosa 36570-900, MG, Brazil; haira.lucio@ufv.br (H.G.L.); rita.lopes@ufv.br (R.d.C.S.O.L.); mariana.juste@hotmail.com (M.J.C.G.); alessan.drasg94@gmail.com (A.d.S.); cmdellalucia@ufv.br (C.M.D.L.); barbara.p.silva@ufv.br (B.P.d.S.); 2Pharmacy and Nutrition Department, Federal University of Espírito Santo, Alto Universitário, City Center, Alegre 29500-000, ES, Brazil; marianagrancieri@gmail.com; 3Embrapa Milho e Sorgo, Rote MG 424, Km 65, Sete Lagoas 35701-970, MG, Brazil; valeria.vieira@embrapa.br

**Keywords:** *Sorghum bicolor* L. Moench, symbiotic meal, gut microbiota, SCFAs, uremic toxins

## Abstract

Background: Chronic kidney disease increases uremic toxins concentrations, which have been associated with intestinal dysbiosis. *Sorghum bicolor* L. Moench has dietary fiber and bioactive compounds, while *Bifidobacterium longum* can promote beneficial health effects. Methods: It is a controlled, randomized, and single-blind clinical trial. Thirty-nine subjects were randomly separated into two groups: symbiotic group (SG), which received 100 mL of unfermented probiotic milk with *Bifidobacterium longum* strain and 40 g of extruded sorghum flakes; and the control group (CG), which received 100 mL of pasteurized milk and 40 g of extruded corn flakes for seven weeks. Results: The uremic toxins decreased, and gastrointestinal symptoms improved intragroup in the SG group. The acetic, propionic, and butyric acid production increased intragroup in the SG group. Regarding α-diversity, the Chao1 index was enhanced in the SG intragroup. The KEGG analysis revealed that symbiotic meal increased the intragroup energy and amino sugar metabolism, in addition to enabling essential amino acid production and metabolism, sucrose degradation, and the biosynthesis of ribonucleotide metabolic pathways. Conclusions: The consumption of symbiotic meal reduced BMI, improved short-chain fatty acid (SCFA) synthesis and gastrointestinal symptoms, increased diversity according to the Chao1 index, and reduced uremic toxins in chronic kidney disease patients.

## 1. Introduction

Chronic kidney disease (CKD) is a clinical syndrome secondary to definitive alteration in the function and/or structure of the kidneys. It is characterized by its irreversibility and slow and progressive evolution, with a high risk of complications and mortality [1]. In 2017, CKD prevalence was between 9.1% and 13.4% in the worldwide population [2]. In addition, CKD is associated with a higher risk of cardiovascular disease, severity, and death [3].

Evidence suggests that CKD causes intestinal dysbiosis, with alterations in gut microbiota composition and intestinal functionality. These effects break the intestinal epithelial barrier and increase intestinal permeability, production, and entry of endotoxins, which favors systemic inflammation. The systemic inflammation promoted by CKD has been associated with a reduction in the populations of beneficial bacteria such as *Lactobacillus* and *Bifidobacterium* and an increase in potentially pathogenic bacteria such as *Escherichia coli* and *Clostridium* spp., microorganisms capable of producing toxins and harmful substances such as p-cresol and indoxyl sulfate, which directly interfere with intestinal health, favoring intestinal dysbiosis, increasing intestinal permeability, and reducing SCFA synthesis [4,5].

In addition, other factors such as the reduction in *Lactobacillus* and *Bifidobacterium*, as well as the low intake of dietary fiber, can also reduce the synthesis of SCFAs, favoring intestinal dysbiosis. The intestinal dysbiosis in CKD includes increased species of the genera Enterobacteriaceae and Pseudomonadaceae of the phylum Proteobacteria, Bacteroidaceae, and Clostridiaceae and decreased species of Lactobacillacea, Prevotellaceae, and Bifidobacteriaceae [4,5,6]. Surprisingly, acting as a vicious cycle, the species that expand in CKD are generally able to induce local and systemic inflammation directly and indirectly [6]. In this context, intestinal dysbiosis is associated with the progression of CKD, including proteinuric renal disorders and associated morbidities, including inflammation, hypertension, and diabetes [6,7].

The consumption of whole grains and cereals promotes healthy intestinal microbiota phenotypes, thus increasing their richness and diversity, as well as the production of short-chain fatty acids (SCFAs) [8]. The BRS 305 sorghum genotype is rich in dietary fiber, polyphenols, condensed tannin, and resistant starch compared to other genotypes of the grain, such as red and white sorghum. Lopes et al. (2018), who used the same genotype and heat treatment for sorghum, revealed that extruded sorghum breakfast cereal was composed of 8.84% of total dietary fiber, of which 8.78% was insoluble fiber and 0.07% was soluble fiber, 71.04% of carbohydrates, 11.26% of proteins, 0.41% of lipids, 1.03% of resistant starch, 1.87% of ash, and 6.57% of moisture. The authors also observed that this extruded sorghum showed 340.33 mg·100^−1^ g of phosphorus, 0.33 mg·100^−1^ g of copper, 1.93 mg·100^−1^ g of zinc, 1.45 mg·100^−1^ g of magnesium, 102.00 mg·100^−1^ g of calcium, 1.45 mg·100^−1^ g of manganese, 5.59 mg·100^−1^ g of iron, and 353.00 mg·100^−1^ g of potassium. In terms of phenolic compounds, the extruded sorghum contains 1.10 ± 0.02 mg of gallic acid equivalent/g of sample of phenolic compounds and 0.71 ± 0.08 catechin equivalent/g of sample of condensed tannins (proanthocyanidins). The antioxidant activity observed in extruded sorghum was 4.68 ± 0.01 µmol trolox/g, the main 3-deoxyanthocyanins present in this sorghum were luteolinidin and 5-methoxyluteolinidin, and the authors detected traces of apigeninidin and 7-methoxyapigeninidin [9].

These compounds are associated with intestinal modulation, as they are non-digestible carbohydrates fermented by gut microorganisms, which increases SCFA synthesis [10]. The beneficial effect of sorghum BRS 305 consumption on health has already been demonstrated. In rodent models, the BRS 305 sorghum whole flour modulated the gut microbiota composition, the abundance of SCFA-producing bacteria, and intestinal morphology [11]. In CKD patients, a symbiotic meal containing BRS 305 extruded sorghum reduced uremic toxins, fecal pH, and urea concentration [12]. 

On the other hand, studies have demonstrated that probiotic supplementation, such as with *Bifidobacterium longum*, isolated or associated with other microorganisms, led to positive changes in the intestinal microbiota, as well as gastrointestinal symptoms, such as increased frequency of bowel movements in healthy subjects [13] or in subjects with persistent gastrointestinal symptoms, such as lactose intolerance [14]. In CKD patients, the administration of a symbiotic meal containing *Bifidobacterium longum* and *Lactobacillus acidophilus* alongside 60 mg of fructooligosaccharides (FOSs) for 60 days improved constipation symptoms and constipation-related quality of life [15]. Another study pointed out that the offer of a low-protein diet (0.6 g/kg/body weight) associated with a probiotic containing *Bifidobacterium longum* 5 × 10^9^ CFU/mL and *Lactobacillus reuteri* 1 × 10^9^ CFU/mL for 60 days reduced blood urea nitrogen and microbiota toxins, including indoxyl sulfate and lipoprotein-associated phospholipase A2 [16]. 

Thus, the present study aimed to investigate the effects of the consumption of a symbiotic meal containing extruded sorghum BRS 305 and Bifidobacterium longum 108CFU/100 mL strain on uremic toxin serum levels, SCFA production, and the gut microbiota composition of CKD patients. We hypothesized that this symbiotic meal may improve gut microbiota diversity, gastrointestinal symptoms, and SCFA production, in addition to reducing the blood level of uremic toxins in CKD patients.

## 2. Materials and Methods

### 2.1. Study Design

This is a controlled, randomized, single-blind clinical trial, conducted for 7 weeks, with CKD patients submitted to hemodialysis for at least 3 months. This study uses data from a subsample of a randomized, controlled, single-blind clinical trial previously conducted by our research group [9,12], since some volunteers donated stool samples for data analysis. The analysis of the intestinal microbiota and its association with markers of intestinal health have not been previously explored, which explains the need for this new investigation study. The participants included in the study were randomly allocated in a 1:1 ratio to receive a symbiotic meal containing extruded BRS 305 whole sorghum plus a probiotic milk containing *Bifidobacterium longum* 2.5 × 10^6^ CFU/mL or pasteurized milk plus extruded corn (Figure 1).

This study was conducted according to the guidelines in the Declaration of Helsinki, and all procedures involving human subjects/patients were approved by the Human Research Ethics Committee of the Federal University of Vicosa, MG, Brazil (protocol number 701.796/2014). It was registered at www.ensaiosclinicos.gov.br under ID number RBR-2d9ny6. Written informed consent was obtained from all subjects/patients.

### 2.2. Participants

The participants were recruited at the Hemodialysis Sector of Hospital São João Batista, located in Viçosa, Minas Gerais, Brazil, between March and June 2015. The first step to recruit volunteers was a meeting with the medical team from the hospital nephrology sector. Then, a conversation with each patient on hemodialysis was conducted to explain to them that a meal containing sorghum and milk would be offered daily for 7 weeks. The patients who agreed to participate in the study were screened to investigate whether they met the eligibility criteria and did not meet any of the non-inclusion criteria. Those who were fit to participate in the study signed the free and informed consent form and were included in the study. Further information about this step is available in Lopes et al. (2018) [9].

Our eligibility criteria were patients of both sexes with CKD who were at least 18 years of age who had been submitted to hemodialysis sessions three times a week in the Nephrology Sector of the Hospital São João Batista, Viçosa, Brazil, for at least three months. The non-inclusion criteria were the presence of auditory deficiency, autoimmune diseases, hepatitis B and C virus infection, implanted catheters, hemodynamic instability, and lactose intolerance or discomfort when consuming milk (Figure 2). The exclusion criteria were the use of antibiotics during the intervention and non-consumption of the symbiotic meal for more than five days (consecutive or not). Participants were characterized by sociodemographic and clinical aspects before the intervention period. The collection of information on sociodemographic and clinical aspects was obtained from medical records and through questions asked in direct interviews, collecting information such as time of disease, associated morbidities, food consumption, measurement of weight and height, and calculation of BMI. They received the meals for a period of 7 weeks, respecting the routine of blood collection at the hemodialysis service.

### 2.3. Randomization, Allocation, and Sample Power

The sequence of allocation and attribution of participants in the two groups was randomly and blindly performed. Therefore, after the randomization, the participants were allocated into two intervention groups—the control group (CG) and the symbiotic group (SG). The randomization was performed by drawing lots, using paper, in which one individual was drawn at random for the CG and another individual for the SG. The sample size required (n = 19/group) was based on the comparison of the means of serum urea levels, considering that it is a relevant variable in renal patients on dialysis. The present study presented 96.68% statistical power (α = 0.05) to detect a 22.9 reduction in serum urea levels, considering the baseline data of our subjects [17].

### 2.4. Raw Material and Meal Preparation

The sorghum grains were provided by Embrapa Milho e Sorgo, Sete Lagoas, MG, Brazil (−19.466672726578615° S, −44.17357630467641° W). The BRS 305 sorghum hybrid is rich in tannins and resistant starch. The grains used in this study were cultivated between April and July 2014, while the corn grains were obtained from the 2013/2014 crop. After harvest, the grains were packed in plastic bags and sent to Embrapa Agroindústria de Alimentos, Rio de Janeiro, Brazil, for the extrusion process. The chemical composition of extruded sorghum and corn was determined by the AOAC methodology (Appendix A) [9,18].

### 2.5. Interventions

All study participants were instructed to follow the usual pattern of diet, physical activity, and lifestyle. Their food intake and intestinal symptoms, as well as marker uremic and inflammatory symptoms, were assessed at the beginning and end of the intervention. Their clinical data, including dialysis time, were collected through the Metabolic Questionnaire adapted from Dixon [19], with multiple-choice questions, to identify the occurrence of gastrointestinal symptoms. In addition, the Bristol scale [20] was applied to verify stool consistency, and the 24 h food recall was used to assess the pattern of food consumption.

The intervention consisted of two groups that received two dairy meals. The dairy meals used in the study were 100 mL of pasteurized milk plus extruded corn, which was supplied to the CG, and 100 mL of pasteurized milk with the probiotic bacteria *Bifidobacterium longum* (Granotec do Brazil S.A) 2.5 × 10^6^ CFU/mL plus extruded sorghum added, supplied to the SG (Appendix A). The beverages were produced weekly at the dairy plant of the Federal University of Viçosa. Milk pasteurized with probiotics was inoculated with a direct vat set (DVS)-type culture from Granotec do Brazil S.A. to present a minimum concentration of viable cells of *Bifidobacterium longum* of 10^8^ UFC/portion (100 mL of milk) [12]. The drinks were packaged in plastic bottles with an aluminum seal and labeled with the following information: date of manufacture, expiration date, and instructions for conservation and consumption. Storage was carried out under refrigeration at 4 ± 2 °C for up to 8 days to preserve the product.

The participants in the control group received a food kit containing pasteurized milk- MP (100 mL) and extruded corn flakes (40 g). The intervention group received a kit with probiotic dairy drink (PDD) (100 mL) containing the Bifidobacterium longum (4 × 108 CFU/100 mL) strain and extruded sorghum flakes (40 g) (Appendix A). The number of extruded cereals offered daily to the volunteers was based on a usual portion of breakfast cereal (40 g) [21]. Two food kits were given to the patients during hemodialysis. One of them should be consumed in the third hour of hemodialysis and the other on the interdialytic day. Patients that could not consume the products in the nephrology sector were instructed to take them home and consume them together on the same day. During hemodialysis, the participants answered a questionnaire about the consumption of the offered meals and the occurrence of adverse effects to assess their adherence to the study protocol and possible complications during the study.

### 2.6. Outcomes

Their feces were collected at the baseline and endpoint of the intervention for the analysis of their gut microbiota and short-chain fatty acids. Stool samples were collected by the participants in sterile bottles and kept at a temperature of −18° until the moment of hemodialysis. The participants transported the containers to the nephrology sector in Styrofoam packaging with ice cubes to maintain the temperature. The samples collected during hemodialysis were aliquoted and stored at −80 °C. The anthropometric measurements and collection of feces and blood samples were carried out in the beginning and at the end of the experiment. 

The present study primarily detected the effects of the interventions on gut microbiota composition and markers related to intestinal health, such as gastrointestinal symptoms, short-chain fatty acid production, and uremic markers. The second outcome refers to food consumption and the effects on biochemical markers related to chronic kidney disease, such as urea and creatinine.

### 2.7. Anthropometric Measures

Body weight was evaluated using an electronic platform scale (Toledo Brazil, Model 2096 PP) capable of handling up to 150 kg and providing measurements with the precision of 50 g. Height was determined using a wall-mounted stadiometer (Alturexata^®^, Belo Horizonte, Brazil). Body mass index (BMI) was calculated as weight in kilograms divided by the square of height in meters (kg/m^2^) and categorized according to the World Health Organization (WHO) guidelines, 2000 [22]. These anthropometric assessments were conducted following the conclusion of the hemodialysis session, thus allowing a 30 min period of hemodynamic stabilization.

### 2.8. Analysis of the Consumption of Macronutrients

The consumption of macronutrients was assessed using a 24 h dietary recall, considering one day of hemodialysis, one interdialytic day, and one weekend. The Dietpro^®^ nutrition software system (version 5i) was used to assess the intake of nutrients.

### 2.9. Uremic Markers

Uremic markers, such as p-cresyl sulfate, indoxyl sulfate, and indole-3-acetic acid, were analyzed and determined in plasma samples by HPLC, according to the method proposed by LOOR et al. (2009) [23]. The method is based on the acidification and centrifugation of the plasma sample. The clear supernatant obtained was injected on a reversed phase HPLC column.

### 2.10. Gastrointestinal Symptoms

The gastrointestinal symptoms were identified by applying multiple-choice questions taken from the Metabolic Questionnaire adapted from DIXON [19], and the shape of the stools was classified according to the Bristol scale [20].

### 2.11. Fecal SCFA Concentrations

Approximately 500 mg of feces was blended while adding 1 mL of ultrapure water to extract the short-chain fatty acids from the fecal samples. Next, the samples were subjected to centrifugation at 12,000× *g* for 10 min at the temperature of 4 °C using a Himac CT 15RE centrifuge from Hitachi (Tokyo, Japan). Subsequently, the supernatants were processed as outlined by Ussar et al. (2015) [24]. The propionic and butyric acids were quantified via high-performance liquid chromatography (HPLC), using a Dionex Ultimate 3000 dual-detector HPLC system (Dionex Corporation, Sunnyvale, CA, USA) coupled with a refractive index (RI) Shodex RI-101 detector (Tokyo, Japan). The following chromatographic conditions were employed: a Bio-Rad HPX-87H column (300 mm × 4.6 mm) (Hercules, CA, USA) equipped with a Bio-Rad Cation H guard column (Hercules, CA, USA), maintained at a column temperature of 45 °C, and a 20 μL injection volume. The mobile phase consisted of concentrated sulfuric acid, EDTA, and ultrapure water, with a flow rate of 0.7 mL/min. The standard curve was calibrated using the following organic acids: acetic, succinic, formic, propionic, valeric, isovaleric, isobutyric, and butyric acid. The standard solutions of these acids were prepared with a final concentration of 10 mmol/L, except for acetic acid, which presented a concentration of 20 mmol/L. The solute levels were determined using standards and a quantitative curve.

### 2.12. Analysis of Intestinal Microbiota

The DNA from stool samples was extracted using the QIAmp DNA stool mini kit for human stool (Qiagen^®^, Venlo, The Netherlands), according to the manufacturer’s protocol. The quality and quantity of the extracted DNA were verified using a µDropTM Plate (Thermo Fisher Scientific, Vantaa, Finland). Integrity and size were measured by agarose gel electrophoresis, and the samples were stored at −20 °C until the time of sequencing analyses. 

The variable regions of the 16S rRNA gene of members of the bacteria domains (V3–V4) were sequenced by the company Argonne National Laboratory^®^ (Lemont, IL, USA), using the MiSeq platform (Illumina, San Diego, CA, USA). Data processing and analysis were performed using the Mothur v.1.40.0 program [25]. The sequences were aligned using the SILVA v.132 16S rRNA gene reference database [26]. The taxonomic classification was carried out using the same database mentioned above. The operational taxonomic unit (OTU) was grouped with a cutoff point of 97% similarity. 

The Chao1, Shannon, and Simpson indices were applied for α-diversity analysis. β-diversity was assessed by principal coordinate analysis (PCoA) based on the Bray–Curtis dissimilarity index and similarity test for non-parametric data (ANOSIM, permutation number = 1000), using the Past software system (HAMMER et al., 2001) [27].

The metagenome functional predictive analysis was carried out using the PICRUSt2 software system. The normalized OTU abundance was identified, and the assigned functional traits were predicted, based on reference genomes, using the Kyoto Encyclopedia of Genes and Genomes (KEGG). The most abundant metabolic processes and significant fold-change differences in functional pathways between experimental groups, adopting unpaired *t*-test control versus symbiotic analysis or paired *t*-test (for beginning- and endpoint group analysis) (α = 95%) using STAMP software version 2.1.3, were plotted.

### 2.13. Statistical Analysis

The dataset was tested for normality by the Kolmogorov–Smirnov test, and parametric data were submitted to ANOVA followed by Tukey’s post hoc test for multiple comparisons. The non-parametric and independent data were submitted to the Kruskal–Wallis test followed by the Mann–Whitney test for multiple comparisons. *T*-tests were applied to compare the baseline and endpoint results of each group. The data were corrected using the FDR (false discovery rate) criterion in the STAMP software. Statistical analyses were performed using GraphPad software version 9.0. Statistical significance was established at *p* < 0.05.

## 3. Results

### 3.1. Baseline Characteristics of Treatment Groups

Thirty-nine subjects completed the study protocol and were included in the analyses; 20 of them were from CG and 19 were from SG. The anthropometric measurements did not differ between the groups at baseline (Table 1). According to the body mass index (BMI), 15% (*n* = 3) of the participants were overweight or obese; 60% (*n* = 12) were eutrophic; and 25% (*n* = 5) were considered underweight. They were 26.81 ± 0.74 years old, with a mean waist circumference of 96.88 ± 1.04 cm. 

### 3.2. Consumption of Macronutrients and Body Mass Index

The consumption of energy, carbohydrates, proteins, and lipids did not differ between the groups during the intervention. Every day, the CG consumed 36.24 ± 13.44 g of lipids, 73.17 ± 30.91 g of protein, and 219.94 ± 92.11 g of carbohydrates. Every day, the symbiotic group consumed 34.94 ± 13.98 g of lipids, 63.45 ± 28.54 g of protein, and 211.19 ± 73.89 g of carbohydrates. On the other hand, the symbiotic consumption reduced the body mass index (BMI) intergroup at the endpoint, with delta equal to −0.079 ± 0.7511 for the symbiotic group and 0.59 ± 1.1204 for the control group (*p* = 0.0479).

### 3.3. Uremic Markers

The serum urea levels were similar inter- and intragroup and did not differ between the groups after the intervention period. Regarding the uremic markers, the symbiotic meal consumption reduced the p-cresyl sulfate and indole-3-acetic acid concentrations intragroup. Considering the delta values, intergroup differences were not observed. The creatinine and urea levels did not change after the intra- and intergroup intervention (Table 2).

### 3.4. Gastrointestinal Symptoms

The consumption of the symbiotic drink increased the evacuation frequency and decreased the gastrointestinal symptoms assessed through the Dixon questionnaire. After the intervention period, 68.4% and 31.6% of the participants allocated to the SG reported having evacuated 5–7 and 2–4 times per week, respectively. On the other hand, 25%, 35%, and 40% of the participants allocated to the CG reported having evacuated 1 time, 5–7, and 2–4 times per week, respectively. In addition, 63.1% and 60% of the participants of the SG and CG groups, respectively, did not present constipation, nausea, heartburn, bloating, intestinal gas, diarrhea, or belching.

According to the Bristol scale, the prevalence of SG participants with a normal consistency of stool, diarrhea, and constipation was 84.2%, 10.5%, and 5.3%, respectively, while for CG, values of 90%, 5%, and 5% were found, respectively (Appendix A).

### 3.5. Fecal SCFA Concentrations

The production of short-chain fatty acids (acetic, propionic, and butyric acid) increased intragroup after the consumption of symbiotic and control meals in both intervention groups when compared to the baseline, with the exception of butyric acid for the CG. The SCFA content did not differ intergroup (Table 3). 

### 3.6. Analysis of Intestinal Microbiota

The sequencing of the 16S rRNA gene from stool samples generated 2,645,395 raw sequences. After filtering and cleaning, 1,890,466 good-quality sequences were obtained. The Good’s coverage obtained in the samples was >99%, which indicates good sequencing coverage. Raw read, filtered read, and normalized read counts per group are provided in the Appendix A. 

The α-diversity, microbial richness estimated by the Chao1 index increased in SG, compared to the baseline (*p* = 0.02) (Figure 3A). However, the Simpson and Shannon indices did not differ between groups after the intervention period (Figure 3B,C).

The β-diversity was assessed at four points. The principal coordinate analysis (PCoA) represented approximately 33.31% and 30.18% of the dissimilarity in bacterial species composition for SG and CG, respectively (Figure 4A,B). At baseline, PCoA represented approximately 30.40% of the dissimilarity in bacterial species composition (Figure 4C). At the endpoint, PCoA represented 30.6% of dissimilarity in bacterial species composition (Figure 4D). The clustering of the bacterial community did not differ between groups at the phyla, class, order, family or genera levels (*p* > 0.05). 

The samples presented 18 phyla, 30 classes, 73 orders, 1141 families, and 373 genera. All groups exhibited eight predominant phyla, including Firmicutes (CG: 73.81 ± 3.89%; SG: 76.71 ± 3.39%), followed by Bacteroidetes (CG: 14.03 ± 3.25%; SG: 11.73 ± 2.33%), Actinobacteria (CG: 6.67 ± 2.20%; SG: 5.59 ± 1.91%), Desulfobacterium (CG: 0.95 ± 0.50%; SG: 0.90 ± 0.48%), and Verrucomicrobia (CG: 0.89 ± 1.05%; SG: 0.92 ± 0.64%) (Appendix A). In the intergroup comparison, the Firmicutes/Bacteroidetes ratio was similar (*p* > 0.05), but in the intragroup comparison, the Firmicutes/Bacteroidetes ratio of CG differed (Appendix A). 

According to the KEGG metabolic pathway analysis, SG increased GPD-D-glycerol-alpha content, D-manno-heptose biosynthesis (*p* = 0.02), L-lysine biosynthesis (*p* = 0.04), methanogenesis from H_2_ and CO_2_ (*p* = 0.04), pantothenate content, coenzyme A biosynthesis I (*p* = 0.04), phosphopantotenate biosynthesis I (*p* = 0.01), pyrimidine deoxyribonucleotide biosynthesis from CTP (*p* = 0.01), and pyrimidine deoxyribonucleotide de novo biosynthesis IV (*p* = 0.01), while reducing the reductive TCA cycle I (*p* = 0.04) metabolic pathway intragroup (Appendix A).

In the comparison of intergroup differences for metabolic pathways by KEGG analysis, the SG reduced 6-hydroxymethyl-dihydropterin diphosphate biosynthesis I (*p* = 0.04), 6-hydroxymethyl-dihydropterin diphosphate biosynthesis III (Chlamydia) (*p* = 0.02), the Calvin–Benson–Bassham cycle (*p* = 0.01), chorismate biosynthesis from 3-dehydroquinate (*p* = 0.04), flavin biosynthesis I (bacteria and plants) (*p* = 0.03), L-lysine biosynthesis I (*p* = 0.01), L-ornithine biosynthesis (*p* = 0.02), N10-formyl-tetrahydrofolate biosynthesis (*p* = 0.04), NAD biosynthesis I (from aspartate) (*p* = 0.01), NAD salvage pathway I (*p* = 0.04), pyrimidine deoxyribonucleotide de novo biosynthesis IV (*p* = 0.008), sucrose degradation III (sucrose invertase) (*p* = 0.01), the superpathway of N-acetylglucosamine, N-acetylmannosamine and N-acetylneuraminate degradation (*p* = 0.03), thiamin salvage II (*p* = 0.03), and thiazole biosynthesis I (*E. coli*) (*p* = 0.03) metabolic pathways (Appendix A).

## 4. Discussion

The present study investigated the effects of the consumption of a symbiotic meal containing extruded BRS 305 hybrid sorghum and extruded corn on the modulation of gut microbiota and the markers associated with uremic parameters in patients with CKD. Symbiotic meal consumption reduced indoxyl sulfate, indole-3 acetic acid (IAA), and p-cresyl sulfate serum concentration intragroup. No differences were observed intergroup. Further, the symbiotic drink ameliorated the intestinal function, enhanced evacuation frequency, reduced gastrointestinal symptoms, and enhanced the number of species at endpoint without altering the Firmicutes/Bacteroidetes ratio or varying genus composition. In this context, the symbiotic meal offered improved the Chao1 index, gastrointestinal symptoms, and SCFA production, in addition to reducing the blood level of uremic toxins in CKD patients.

Symbiotic meal consumption increased acetic, propionic, and butyric acid levels intragroup. Probiotics like *Bifidobacterium longum* helped to re-establish a healthy gut microbiota by enhancing the growth of beneficial bacteria and reducing the levels of pathogenic bacteria and uremic toxins. The sorghum, a source of dietary fiber, acts like a prebiotic, providing the necessary nutrients to support the growth and activity of these probiotics, promoting colonic fermentation, resulting in an increase in SCFA synthesis and concentration. In this context, the combined action of probiotics and prebiotics in a symbiotic meal can enhance the production of short-chain fatty acids (SCFAs), helping to maintain gut barrier integrity and reduce inflammation. This symbiotic approach not only improves gut health but also potentially mitigates CKD progression by reducing systemic inflammation and uremic toxin levels [28,29]. In the intestinal environment, SCFAs provide energy for colonocytes, thus modulating their proliferation, differentiation, and the inhibition of pathogenic bacteria growth, in addition to strengthening the intestinal barrier, reducing luminal pH and intestinal permeability, and improving the immune function of CKD patients [30,31]. 

Symbiotic meal consumption reduced uremic toxins, such as indole-3-acetic acid (IAA) and p-cresyl sulfate (p-CS). Uremic toxins are usually increased in CKD patients. They can alter the intestinal microbiota and promote dysbiosis by increasing intestinal permeability [32] and pH, which facilitates the growth of pathogenic microorganisms [33,34] and favors the progression of CKD [34,35]. However, the symbiotic meal increased SCFA content, which probably inactivated the bacterial families associated with the production of uremic toxins, such as p-CS and IAA. Further, probiotic intake can increase acetic acid production, to which Bifidobacteria are mainly associated [36]. In addition, no increase in the number of lactic acid bacteria was observed in our study, since the production of butyric and propionic acids increased and may be associated with the amount of dietary fiber, resistant starch [37], and other bioactive compounds present in extruded sorghum [12], such as 3-deoxyanthocyanins and condensed tannins. It is known that butyric acid improves the intestinal barrier function and inhibits the generation of p-CS and the activation of marker inflammation [29,38]. In addition, the consumption of resistant starch was associated with reduced uremic toxin serum levels [35,38].

The bioactive compounds, such as condensed tannins and 3-deoxyanthocyanins and dietary fibers from sorghum, are related to the lower digestibility of the cereal when compared to corn [39], which can favor BMI reduction intergroup. Although no differences in intergroup food intake were observed in the present study, the administration of Bifidobacterium longum has been associated with other microorganisms and weight loss in obese individuals due to reduced microbiome lipopolysaccharides and a consequent increase in satiety [40,41].

In our study, the presence of dietary fiber and resistant starch through the consumption of the symbiotic meal increased the frequency of bowel movements. The fermentation of soluble fiber is associated with increased SCFA production, while the fermentation of insoluble fiber accelerates intestinal transit and increases the fecal bolus [32,35], which improves stool consistency and reduces intestinal constipation [42]. Beneficial effects on gastrointestinal symptoms were observed by Cruz-Mora et al. (2014) in CKD patients receiving a probiotic and inulin [43]. In addition, symbiotics containing *Bifidobacterium longum* are associated with reduced gastrointestinal symptoms, including constipation and ameliorated life quality, according to the Patient Assessment of Constipation Quality of Life (PAC-QOL) questionnaire [16]. Therefore, this symbiotic meal can promote a favorable intestinal condition.

In CKD patients, dysbiosis is frequent and characterized by reduced beneficial commensal bacteria and increased uremic toxin-producing bacteria [44,45]. In our study, the symbiotic meal improved α-diversity, thus increasing the abundance of microbial genera. Other indices related to α-diversity, such as Shannon and Simpson indices, indicated no changes in the number or dominance of species. The β-diversity supported this result, without inter- or intragroup differences. Other factors, such as stress, aging [46], obesity/fat accumulation [47], diet quality [48], and occurrence of other diseases, can alter gut microbiota composition [49,50], which may hinder the observation of positive changes during 7 weeks of symbiotic meal intervention. In agreement with our results, the consumption of a diet rich in dietary fiber provided by whole grains did not change α-diversity or β-diversity [51,52].

The increase in the Chao1 index without changes in Shannon and Simpson indices suggests that the symbiotic meal had a specific impact on the alpha diversity of the intestinal microbiota of chronic kidney disease patients on hemodialysis, increasing the number of rare or less abundant species—that is, increasing their wealth. The results obtained for the Shannon and Simpson indices suggest that although more species may have been introduced or flourished due to treatment with the symbiotic meal, the abundance of the dominant species was not significantly changed. Therefore, the symbiotic meal may have contributed to a greater diversity of rare species in the intestinal microbiota without modifying the general structure of the microbial community [53].

The KEGG analysis demonstrated that the symbiotic meal improved pathways related to energy metabolism, amino sugar metabolism [54], essential amino acid production and metabolism [55], degradation of sucrose [56], and the biosynthesis of ribonucleotides [57]. The predictive effects observed in amino acid and amino sugar pathways are related to improved immune function, oxidative stress, and immune response [11]. Further, the predictive analysis revealed an increased L-ornithine pathway, an intermediate compound in L-arginine biosynthesis, which, in turn, is used to synthesize glutamate. Glutamate is an amino acid with beneficial effects on intestinal barrier function, which reduces the entrance of endotoxins [58,59]. 

The main limitations of this study included CKD patients with long hemodialysis treatment (more than 50 months, on average, for both groups), the time of the intervention, only one type of probiotic used, and the lack of control groups with sorghum and *Bifidobacterium longum*. Thus, our study revealed that the consumption of symbiotic meal with extruded BRS 305 grains associated with *Bifidobacterium longum* was effective in improving gastrointestinal symptoms, stool consistency, and SCFA production, in addition to reducing uremic toxins, such as p-CS and IAA, possibly favoring enterocyte proliferation. Despite the favorable results of this symbiotic meal consumption, further studies are necessary to evaluate the long-term effect of symbiotic meal consumption on the biochemical (creatinine, urea, and uric acid) and intestinal health (gut microbiota composition, intestinal permeability, feces pH, and stool consistency) parameters of CKD patients undergoing hemodialysis.

## 5. Conclusions

The symbiotic meal containing extruded sorghum BRS 305 associated with *Bifidobacterium longum* was able to improve SCFA production, reduce uremic toxin serum levels, and decrease BMI in CKD patients. Furthermore, the beverage increased bacterial richness and metabolic pathways related to energy metabolism and the biosynthesis of amino acids. Therefore, the symbiotic meal improved intestinal and systemic health status in chronic kidney disease patients.

## Figures and Tables

**Figure 1 nutrients-16-01852-f001:**
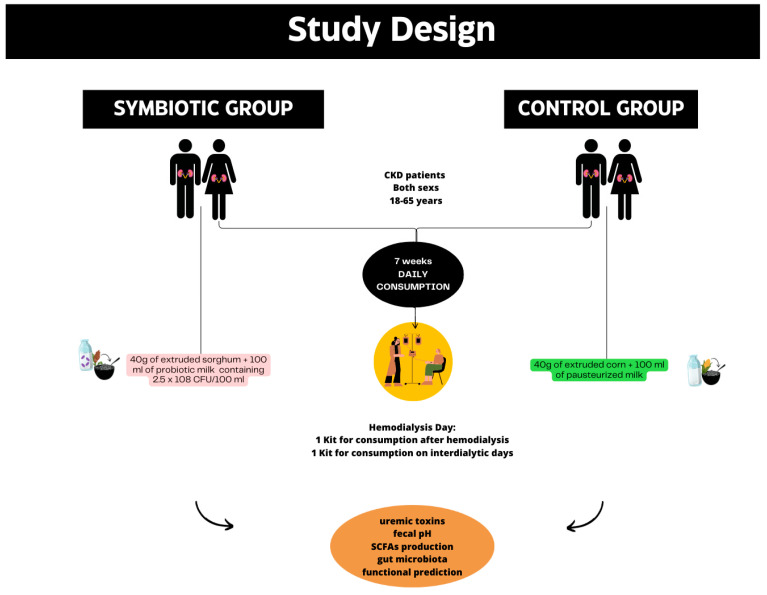
Study design of research.

**Figure 2 nutrients-16-01852-f002:**
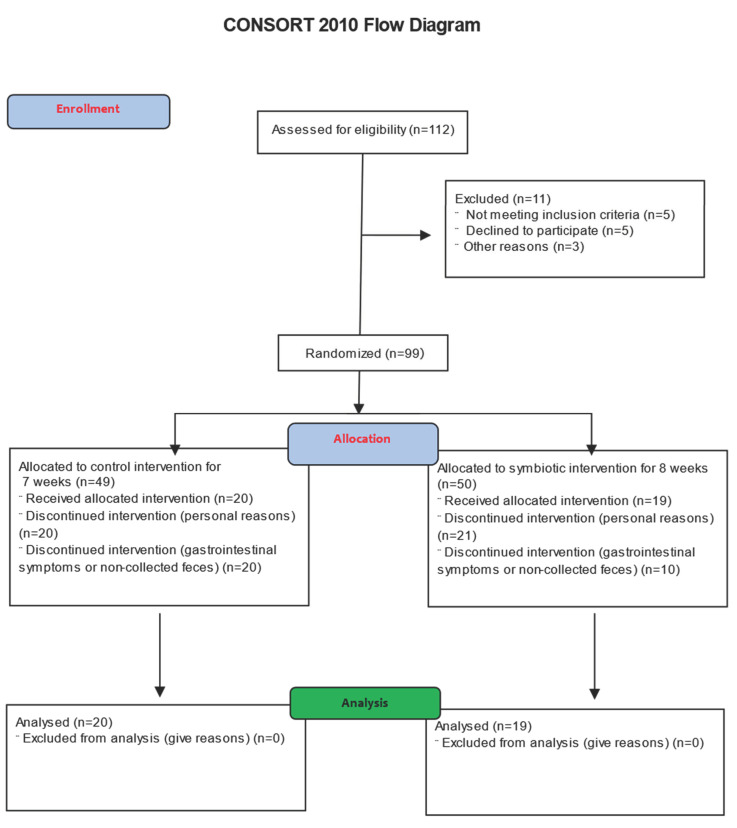
Consort flow diagram of study design of intervention protocol.

**Figure 3 nutrients-16-01852-f003:**
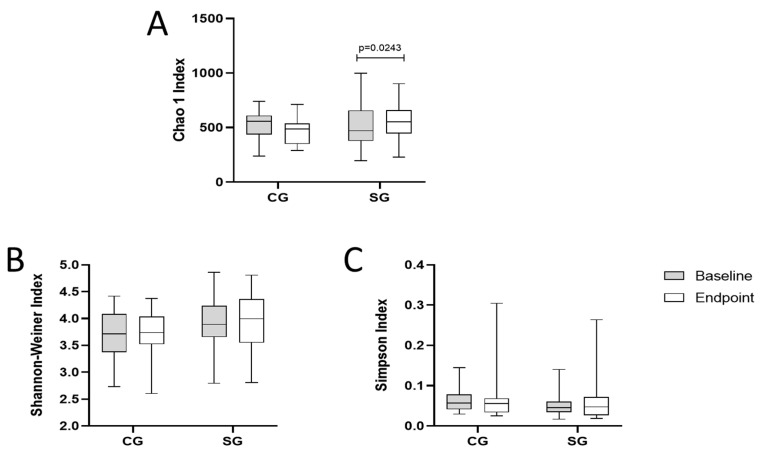
Effect of extruded sorghum BRS 305 plus Bifidobacterium longum on α-diversity index. (**A**) Chao1 index at baseline and endpoint, (**B**) Shannon–Weiner index at baseline and endpoint, (**C**) Simpson index at baseline and endpoint. CG: control group; SG: symbiotic group. The data were subjected to a paired t-test or unpaired *t*-test (α = 0.05) in GraphPad version 9.0.

**Figure 4 nutrients-16-01852-f004:**
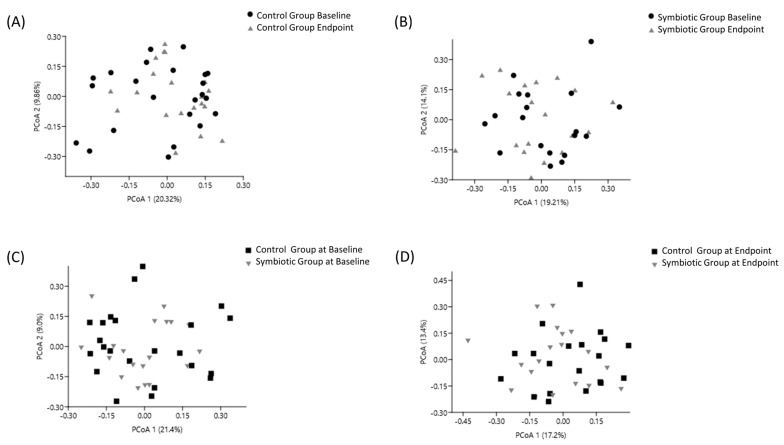
Effect of extruded sorghum BRS 305 plus *Bifidobacterium longum* on the β-diversity estimated by principal coordinate analysis (PcoA), based on the Jaccard similarity distance of gut microbial communities in chronic kidney disease patients. (**A**) PCoA of the symbiotic group at baseline and endpoint; (**B**) PCoA of the control group at baseline and endpoint; (**C**) PCoA of the control and symbiotic groups at baseline; (**D**) PCoA of the control and symbiotic groups at endpoint. Permutational multivariate analysis of variance (PERMANOVA) was conducted using the STAMP software system version 2.0.2 considering α = 5%.

**Table 1 nutrients-16-01852-t001:** Baseline characteristics of the study participants.

Variables	Symbiotic Group	Control Group	*p* Value
Subjects (*n* = 39)	19	20	-
Sex	Man: 12 Woman: 7	Man: 15 Woman: 5	-
Age (years)	62.85 ± 11.74	64.22 ± 9.68	0.68
Body weight (kg)	66.06 ± 10.79	59.20 ± 9.89	0.05
BMI (kg/m²)	25.96 ± 4.68	23.08 ± 3.09	0.05
HD time (months)	59.60 ± 72.79	50.83 ± 55.05	>0.99

BMI: body mass index; HD: hemodialysis. The data were subjected to unpaired *t*-test or Mann–Whitney test at 5% probability in GraphPad prism version 9.0.

**Table 2 nutrients-16-01852-t002:** Uremic marker blood concentrations in CKD patients who received symbiotic or control meal by 7 weeks.

Variables	Symbiotic Group		*p*¹ Value	Control Group		*p*¹ Value	Delta *p* Value
	Baseline	Endpoint		Baseline	Endpoint		
IS (mg/dL)	140.46 ± 70.85	115.95 ± 55.65	0.1297	151.94 ± 61.13	147.74 ± 51.36	0.7075	0.1758
IAA (µg/L)	24.21 ± 13.73	18.19 ± 10.67	0.0030	18.62 ± 13.44	15.62 ± 4.91	0.9764	0.3891
p-CS (mg/L)	386.47 ± 197.99	241.13 ± 99.79	0.0001	289.21 ± 245.62	295.02 ± 127.18	0.065	0.3524
Urea Creatinine	37.15 ± 16.70 8.31 ± 3.23	37.80 ± 12.42 8.67 ± 2.60	0.8835 0.1011	43.70 ± 32.88 8.44 ± 3.30	33.09 ± 16.35 9.01 ± 3.93	0.1482 0.1530	0.2175 0.7016

IS: indoxyl sulfate; IAA: indole-3-acetic acid; p-CS: p-cresyl sulfate. Values expressed as mean ± standard deviation (SD). Data were subjected to an unpaired *t*-test or a Mann–Whitney test at 5% probability in GraphPad prism version 9.0. *p*¹ means the difference comparing baseline and endpoint of each intervention group.

**Table 3 nutrients-16-01852-t003:** Short-chain fatty acid fecal concentrations in patients with CKD who received symbiotic or control meal by 7 weeks.

Variables	Symbiotic Group		*p*¹ Value	Control Group		*p*¹ Value	Delta *p* Value
	Baseline	Endpoint		Baseline	Endpoint		
Acetic acid	3.71 ± 1.76	7.00 ± 2.60	<0.0001	4.91 ± 2.07	7.95 ± 3.98	0.0007	0.1758
Propionic acid	2.41 ± 2.04	6.38 ± 3.73	<0.0001	3.05 ± 2.91	5.95 ± 3.71	0.0050	0.3891
Butyric acid	2.26 ± 1.66	3.87 ± 2.38	0.040	2.57 ± 1.91	4.42 ± 4.10	0.0636	0.3524

Values expressed as mean ± standard deviation (SD). Data were subjected to an unpaired *t*-test or a Mann–Whitney test at 5% probability in GraphPad prism version 9.0. *p*¹ means the difference comparing baseline and endpoint of each intervention group.

## Data Availability

The data presented in this study are available upon request to the corresponding author. The data are not publicly available due to the fact that they are available within an internal database of the research institution; therefore, they cannot be made publicly available.

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
