# Peer review of "A Symbiotic Meal Containing Extruded Sorghum and Probiotic (Bifidobacterium longum) Ameliorated Intestinal Health Markers in Individuals with Chronic Kidney Disease: A Secondary Analysis of a Subsample from a Previous Randomized and Controlled Clinical Trial"

_nutrients, 2024, doi:10.3390/nu16121852_

Round 1

Reviewer 1 Report

Comments and Suggestions for Authors

See the comments on the file attached. Obrigado.

Author Response

Thank you very much for the opportunity to revise the article “A symbiotic meal containing extruded sorghum and probiotic (Bifidobacterium longum) ameliorated intestinal health markers in individuals with chronic kidney disease: a secondary analysis of a subsample from a previous randomized, and controlled clinical trial”.

The article is very interesting and well written. Authors concluded that the consumption of symbiotic meal reduced BMI, improved short chain fatty acid synthesis and gastrointestinal symptoms, increased the diversity by Chao1 Index and reduced uremic toxins of chronic kidney disease patients.

I have just a few comments.

1) In the introduction, please describe in more details how CKD causes intestinal dysbiosis, with alterations in gut microbiota composition and intestinal functionality.

Author’s response: Thank you for comment! The information was added in the introduction. Please, see lines 46-55.

2) In the methods, please add the information about patients’ medical history.

Author’s response: Thank you for the comment! The information about patients was included in the section Participants of methods. Please see lines 149-154.

3) In the discussion, please describe in more details how symbiotic meal consumption increased acetic, propionic, and butyric acid intragroup.

Author’s response: Than you very much for this comment! The information was added to discussion in the second paragraph! Please check the lines 424-433.

Reviewer 2 Report

Comments and Suggestions for Authors

Dear Redactors,

Thank you very much for the opportunity to revise the article “A symbiotic meal containing extruded sorghum and probiotic (Bifidobacterium longum) ameliorated intestinal health markers in individuals with chronic kidney disease: a secondary analysis of a subsample from a previous randomized, and controlled clinical trial”.

The article is very interesting and well written. Authors concluded that the consumption of symbiotic meal reduced BMI, improved short chain fatty acid synthesis and gastrointestinal symptoms, increased the diversity by Chao1 Index and reduced uremic toxins of chronic kidney disease patients.

I have just a few comments.

In the introduction please describe in more details how CKD causes intestinal dysbiosis, with alterations in gut microbiota composition and intestinal functionality.

In the methods please add the information about patients medical history.

In the discussion, please describe in more details how symbiotic meal consumption increased acetic, propionic, and butyric acid intragroup.

Thanks

Author Response

We would like to take this opportunity to express our sincere gratitude to the Reviewers for their careful and constructive reviews in our manuscript "A symbiotic meal containing extruded sorghum and probiotic (Bifidobacterium longum) ameliorated intestinal health markers in individuals with chronic kidney disease: a secondary analysis of a subsample from a previous randomized, and controlled clinical trial". The reviewer’s recommendation improved our manuscript. We listed the original comments followed by our response to the comment. The revised manuscript follows with all the modifications highlighted (in red) in the text.

REVIEWER #2:

1) It is fantastic that you submitted your clinical study to the Brazil Clinical Assays website, but why not on ClinicalTrials.gov? I encourage you.

Author’s response: Thank you for the highlight! We appreciate your suggestion! We usually submit our studies to the Ethics Committees of our Research Institution (CEPE-UFV) and the website rehearsalsclinicos.gov.br is recommended in Brazil for studies with humans. The protocol number of acceptance on CEPE – UFV is 701.796/2014 and the ID number of register in  www.ensaiosclinicos.gov.br is RBR-2d9ny6.

2) Line 58. BRS 305 sorghum genotype. I encourage you to add some literature that explains the main genetic characteristics of this sorghum variety, and food analysis performed by you or the company that produces the batch/batches you have been using.

Author’s response: Thank you for the comment. The information about the BRS 305 sorghum nutritional composition was added in the introduction. Please, see lines 67-80.

3) Line 64. Please write the species names in italics.

Author’s response: Thank you very much for your comment. We agree with the reviewer. We wrote the species names in italics. Please, see line 89.

4) Line 72. Correct the scientific notation of the cfu numbers.

Author’s response: Thank you very much for your comment. We agree with the reviewer. We wrote the scientific natation the de CFU number correctly. Please, see lines 97-98.

5) Line 91. Italics.

Author’s response: Thank you very much for your comment. We agree with the reviewer. We wrote the species names in italics. Please, see line 116.

6) Figure 2. in Allocation section, the right apart, appears a “0” mistyped.

Author’s response: Thank you for the comment! The figure was updated and replaced in the text. Please check the figure between 158 and 159 lines.

7) Line 160. In italics

Author’s response: Thank you very much for your comment. We agree with the reviewer. We wrote the species names in italics. Please, see lines 193-197.

8) Line 165. Correctly write the scientific notation of the number.

Author’s response: Thank you very much for your comment. We agree with the reviewer. We wrote the scientific notation the CFU number correctly. Please, see lines 116 and 193.

9) Please, as supplementary material, submit a certificate analysis from you research group or from Granatec do Brazil in order to demonstrate that there are exactly 108 cfu/100 ml. As quality control of your experimental design, I mean.

Author’s response: Our research group carried out a viability analysis of the microorganisms present in probiotic milk, to verify the presence of 108 cells per portion of probiotic milk. The analysis report is attached to the supplementary material as Supplementary Figure 1.

10) Also please, submit a certificate of analysis of your kit because you specify a narrow cfu: 4x108 cfu/100 ml. In addition, some kind of validation you must do to demonstrate that the primary package and storing up to 8 days, you still have 4x108 cfu/100 ml. It is critical to ensure this in your design study.

Author’s response: As answered in item 9, the cell viability report of probiotic milk after storage for 8 days is now attached to the supplementary material of this paper. The Embrapa laboratory, where the analysis carried, has an analysis accredited by ISO 17:025.

11) Why you use a novel software such as PICRUST2 and an ancient taxonomy classifier (SILVA v132 instead of v138)?

Author’s response: Thank you very much for the comment! The SILVA v132 taxonomic classifier was used in a protocol for analyzing the composition of the intestinal microbiota previously proposed when conducting analyzes carried out with the mothur software. However, we will consider updating SILVA v138 for future analyzes carried out by our research group.

12) Table 1. Homogenize the number of decimals used for p-values (0.05 and 0.051).

Author’s response: Thank you very much for your comment. We homogenized the numbers of decimals used. Please, see Table 1, line 311.

13) Table 2. What do you mean in the last column (delta p value)?

Author’s response: Thank you very much for your comment. Yes. It is mean delta p value. The delta p value corresponds to the statistical analysis between groups based on the difference observed between the baseline and the end point for each variable. We modified the write. Please, see Table 2, line 329 and Table 3, line 351.

14) Table 3. Why only 3 SCFAs were reported. Nothing more detected? It is strange.

Author’s response: Thank you for the highlight! The short-chain fatty acid analysis was conducted on total fatty acids and butyric, acetic and propionic acids. The analysis was conducted on this way because these three fatty acids are the predominantly SCFAs produced and involved in the maintenance and integrity of intestinal health. The other short-chain fatty acids produced have a lower concentration when compared to the butyric, acetic and propionic acids.

15) Line 460. Specie in italics. I don’t like to highlight only the species because apart from probiotics you administered sorghum BRS305. Maybe you can highlight the entire sentence

Author’s response: Thank you very much for your comment. We agree with the reviewer. We preferred remove the highlight of species. Please see line 511.

16) In the Discussion section, you could discuss the alpha-diversity in more depth and why Simpson and Shannon give different trends from Chao1. Remember which characteristics measure each index.

Author’s response: Thank you very much for the comment! We agree with this point and the information was added to Discussion section. Please see lines: 480-488.

17) I strongly recommend including a differential abundance analysis if you detected some genus differently present in the two groups. That will enrich your manuscript and your findings. You have done the functional analysis, but before there is the differential abundance analysis (such as Aldex2, LefSe…).

Author’s response: The KEGG functional prediction analysis revealed no significant differences between groups for kingdom-to-genus classifications. Therefore, other analyses, such as LefSe, become unfeasible and were therefore not applied.

Round 2

Reviewer 1 Report

Comments and Suggestions for Authors

Accept in the present form.